# Development and Validation of Recreational Sport Well-Being Scale

**DOI:** 10.3390/ijerph19148764

**Published:** 2022-07-19

**Authors:** Lu-Luan Pi, Chia-Ming Chang, Hsi-Han Lin

**Affiliations:** 1Department of Recreation and Sports Management, University of Taipei, Taipei 111036, Taiwan; peggy@utaipei.edu.tw; 2Department of Physical Education, Health & Recreation, National Chiayi University, Chiayi 621302, Taiwan; gr5166@yahoo.com.tw; 3Department of Tourism and Leisure, Lunghwa University of Science and Technology, Taoyuan 333326, Taiwan

**Keywords:** life satisfaction, positive emotion, physical and mental health, family flourishing

## Abstract

The purpose of this study is to develop the “Recreational Sport Well-being Scale”, which will be used to investigate the subjective recreational sport well-being individuals’ experience after participating in recreational sports. The study participants were Taiwanese who were over 20 years old and participated in recreational sports. Four sets of samples and 4050 questionnaires in total were collected. Using exploratory factor analysis, four factors were extracted from the scale –life satisfaction, physical and mental health, family flourishing, and positive feelings. Confirmatory factor analysis indicated that the scale’s overall goodness of fit, convergent validity, and composite reliability all passed the thresholds. The results of cross-validation indicated that the model passed configural invariance, metric invariance, covariance invariance, and error variance invariance, which suggested that the scale has cross validity. Nomological validity analysis was conducted, showing that Recreational Sport Well-being Scale is nomologically valid since it is positively correlated to Subjective Health Scale. Test-retest reliability analysis suggested that the test results were stable when a retest was carried out two weeks later. The developed “Recreational Sport Well-being Scale” is highly reliable and valid and can be applied to measure future recreational sports participants’ well-being.

## 1. Introduction

Well-being is individuals’ response to and comments on their life experience [1], the overall appraisal of the felt positive and negative feelings, and the degree of life satisfaction [2]. It is a subjective cognitive appraisal and emotional feelings about life. Cognition is satisfaction comments on the overall life. The feeling is the emotional response to positive and negative feelings. It is also a joyful and pleasant experience [3,4]. Well-being is the individuals’ perceived feeling of meaning and pleasure. An individual with more well-being is usually a mentally healthy person. With more positive and less negative emotions, an individual’s well-being level will be raised [5].

The concept of well-being originated from two ancient philosophical concepts—hedonic proposed by Aristippus and eudaimonia, also called self-actualization [6], proposed by Aristotle. Stemming from the concepts mentioned above, well-being is elaborated as follows:

Subjective well-being (SWB) is based on hedonic. Well-being is happiness. The purpose of life lies in the need for satisfaction and to escape pain. Well-being consists of individuals’ subjective happiness. Common measurement structure comprises dimensions like life satisfaction, positive emotion, and negative emotion [3].

Eudaimonic well-being (EWB) is based on self-actualizations. It emphasizes that the key to well-being lies in self-actualization. Purposes in life consist of the pursuit of meaning as well as self-actualization. In addition, authentic pleasure can be gained when an individual’s potential is fully developed. The common measurement structure consists of self-acceptance, positive relationships with others, autonomy, environmental mastery, and personal growth [7].

According to Huta and Ryan [8], hedonia and eudaimonia occupy both overlapping and distinct niches within a complete picture of well-being, and their combination may be associated with the greatest well-being.

In terms of social well-being, two dimensions are commonly employed to measure social well-being –social adjustment and social support. Social adjustment is composed of satisfaction of interpersonal relationships, the performance of social life, and environmental adaptation. Social support consists of two factors in an individual’s interpersonal network –the number of people an individual contacts and how satisfied an individual feels with those people [9]. Social well-being stems from social tasks an individual fulfills in their social organizations or communities. Common measurement structure is composed of social actualization, social coherence, social contribution, social acceptance, and social integration [10].

Emotional well-being is related to mental health, which is a state of positive feelings and positive functioning in life. Emotional well-being is the interpretation of positive feelings. It is also considered to be an individual’s awareness and evaluation of their own emotional state in life. The measurement structure contains four dimensions, namely positive emotion, negative emotion, perceived satisfaction with life, and perceived avowed well-being [11].

The other factor, flourishing, contains five elements—positive emotion, engagement, relationship, meaning, and accomplishment [12]. There were scholars proposing another measurement structure that is composed of five dimensions—physical and mental health, well-being and life satisfaction, meaning and purpose, character and virtue, and close social relationships [13].

According to United Nations’ “The 2018 World Well-being Report”, Taiwanese’ well-being was ranked as the 26th place among 156 countries from 2015 to 2017. The well-being indexes include the gross domestic product (GDP), social support, expected physical health, freedom of making choices in life, generosity, and evaluation of government corruption [14]. However, in the 2020 World Well-being Report, Taiwan ranked 47th place in the world. Obviously, the ranking went down. Therefore, the government needs to care for people’s well-being. As we know, well-being is related to public health. In general, well-being is a measure of people’s overall life satisfaction and contributes to health. In the field of public health, exercise is well-recognized as an important factor that contributes to both physical and mental health. Both physical and mental health contributes to well-being.

Exercises not only enhance well-being but also is the best predicting factor for well-being [15,16]. Therefore, regular engagement in recreational sports can contribute to some extent to well-being. Recreational sports refer to physical activities individuals engage in after work in their free time, which enable them to enjoy the fun of sports, increase physical activities and attain the goal of health [17]. Recreational sports vary from children’s play to various kinds of sports, all of which are valuable recreational sports experiences and can offer participants inner self-satisfaction [18]. Participating in recreational sports can facilitate health, so the public places increasing emphasis on recreational sports and tend to arrange some recreational sports activities. Recreational sports can also strengthen our body and our brain will secrete many sorts of chemicals and neurological media, generating serotonin, which helps us calm emotions, inhibit impulses, and raise our self-esteem [19]. Endorphin, a natural pain reliever, will also be secreted to relieve pain and enhance mental pleasure. It also works well in fighting against depression [20]. Epinephrine will also be generated; it helps awaken the brain, makes the brain function well, and strengthens our memory [21]. Dopamine, related to desire, excitement, happiness, and well-being, will also be secreted so that an individual can be in a good mood [21]. Through constant regular exercise, an individual can gain physical health and mental pleasure, have more positive emotions, and ones well-being level will be raised.

With advanced AI technology, people cling to sedentary lifestyles, which causes serious physical and mental health problems. Therefore, focusing on exercise and health can bring well-being and draws global attention. Meanwhile, research personnel from the fields of psychology, sociology, and recreation, began to take great interest in studies on factors affecting well-being [22]. Gradually, more and more studies focused on the field of well-being. Studies, as well as the survey United Nations conducted, have long shown that well-being is of great importance when it comes to human health. However, in the field of sports, well-being studies usually adopted a subjective well-being scale as a research tool to measure a person’s well-being induced by recreational sports involvement. From previous studies, we believe that recreational sports’ involvement can contribute to both SWB and EWB. Therefore, we see the need for a specific measure that can be used to measure exclusive well-being, which includes both SWB and EWB resulting from recreation sports’ involvement. Hence, this study was designed to develop a sport-related well-being scale to provide a measurement tool for researchers to assess the global well-being induced by recreational sports involvement.

## 2. Materials and Methods

### 2.1. Sampling and Administration

The scope of the study included sports venues, parks, citizen sports centers, communities colleges established and managed by the government in both Taipei and New Taipei City, recreational sports courses, events and games held by profit or nonprofit sports associations or corporations, and members of sport-related clubs in colleges.

Previously, scholars adopted an inductive method and categorized recreational sports into nine types—balls, fitness, outdoor, meetup, parent and kid, defense, dance, health facilitation, and folk sports [17]. Based on the nine types mentioned above, this study adopted convenience sampling and carried out a 2-stage questionnaire survey. In the first stage, 450 questionnaires were collected from the nine types of recreational sports participants; therefore, 4050 questionnaires were collected in all. Valid questionnaires were then divided into 3 groups: Group 1 (n = 1350), Group 2 (n = 1350), and Group 3 (n = 1350). Two weeks later, 40 of the participants took part in the 2nd-stage survey. For the age variable, this study referred to the human life cycle and divided life into six phases, including infancy (0–5 years old), childhood (6–12 years old), adolescence (13–25 years old), maturity (26–50 years old), midlife (51–64 years old), and gerontal (above 65 years old) [17]. When applying for IRB, this study set the participants’ age at over 20 years old; as a result, this study defined adolescence as 20 to 25 years old, maturity as 26–50 years old, midlife as 51–64 years old, and gerontal as above 65 years old.

This study applied for and passed the Institutional Review Board evaluation (No: IRB-2019-056).

### 2.2. Statistical Analysis

The questionnaire of this study included three parts: demographic information, recreational sport well-being scale, and perceived health scale. Demographic information has four items: gender, age, marital state, and education level. Both recreational sport well-being scale and perceived health scale adopted the Likert 5-point scale. 5, 4, 3, 2, 1 point respectively stands for “strongly agree,” “agree,” “neither agree nor disagree,” “disagree,” and “strongly disagree.” The higher point means a greater extent of agreement.

The scale development process of this study referred to research designs of previous studies [23,24,25] and can be divided into six stages.

Stage 1: Questionnaire Development

Scale items were collected in this stage. The researcher collected and read relevant literature concerning recreational sports well-being and drafted a recreational sports well-being scale (please check Appendix A for item details). Scholars were invited to examine the content validity of the scale. After the expert evaluation, an item analysis was carried out with Group 1 participants (n = 1350). A *t*-test was conducted to make sure that each item could be discriminated between. Pearson product-moment correlation analysis was employed to ensure every item is homogeneous to the overall scale [26].

Stage 2: Exploratory Factor Analysis (EFA)

The main purpose of this stage was to construct the factor structure of the recreational sport well-being scale and to test the reliability of the scale with exploratory factor analysis. Group 1 participants were used as samples (n = 1350) and for EFA, the principal component method was employed to extract factors and the Promax rotation technique was adopted to keep the items whose eigenvalue is more than 1. The threshold for factor loading is greater or equal to 0.50 and meets the requirement of unidimension [27].

Stage 3: Confirmatory Factor Analysis (CFA)

In this stage, the factor structure of the recreational sport well-being scale established through EFA was used as the theoretical model and the model was tested with confirmatory factor analysis. Group 2 participants were used as samples (n = 1350) for CFA. Based on the CFA principles previously proposed by scholars, CFA criteria of this studies are as follows [27]: 1. The goodness-of-fit of the measurement model: generally, there are several standards: S-B χ^2^/df < 5, SRMR < 0.10, RMSEA < 0.08, CFI > 0.90, NNFI > 0.90, GFI > 0.90. If all the standards are met, the model has good goodness-of-fit. 2. Convergent validity test: if the standardized factor loadings of the observed variable in the advised model are higher than 0.50, the model has good convergent validity. 3. Discriminant validity: for each factor in the scale, the square root of the average variance extracted should be over 0.50. Moreover, the square root of the average variance extracted should be higher than the factor’s covariance with other factors in the scale. If the requirements mentioned above are met, the scale has discriminant validity. Last but not least, the reliability of the scale will be tested; if the composite reliability is higher than 0.70, the scale has acceptable reliability.

Stage 4: Cross-Validation

In stage 3, the recreational sport well-being model tested with Group 2 data can be used as the based model. To understand whether the based model can be applied to different samples, this study will use Group 3 as cross-validation samples (n = 1350). The analysis process includes examining if there is an equivalence in factor loading, structure coefficient, and measurement residual. If the model passes the equivalence test, the scale has cross validity.

Stage 5: Nomological Validity

Nomological validity refers to the theoretical relationship among important concepts [28] and serves as a key to psychometric measurement. It is also an effective step to test the validity of a scale. The main purpose of the stage is to test the relationship between the score measured on the recreational sport well-being scale and the criterion. A stronger relationship reflects the authentic content of the scale [29]. Previous studies revealed that well-being is correlated to perceived health [30,31,32], so perceived health can be taken as the criterion of well-being. The perceived health scale in this study referred to other perceived health scales previously developed [32], which contain 6 items and can serve as the criterion tool when testing the nomological validity of the study. When the correlation coefficient between recreational sport well-being and perceived is higher than 0.60 and reaches a significant level, the scale has nomological validity. Both Group 2 and Group 3 will be used as the samples (n = 2700) as both groups will fill in the perceived health scale.

Stage 6: Test-Retest Reliability

The main purpose of this stage is to carry out two tests with the same group of participants and to conduct a correlation analysis between two test scores. If the correlation analysis reaches a significant level and the correlation coefficient is higher than 0.60, the scale has test-retest reliability. It is advised that the interval between the two tests should be at least two weeks [26]. In this stage, the items tested are the ones that passed confirmatory factor analysis, and the 40 participants will be selected as samples. They will fill in the questionnaires twice and the second test will conduct two weeks after the first test.

## 3. Results

### 3.1. Description of Sample

This study collected four sets of samples, which were named Group 1 to Group 4. The total of samples was 4090. The gender ratio is balanced. Males take up 49.5% and females 50.4%. In terms of age, 51.8% of them are in the phase of maturity (26 to 50 years old). As for marital status, 58.5% of them are not married. 66% of them received a college education. The demographic information of the samples is shown in Table 1.

### 3.2. Stage 1 Questionnaire Development

This study adopted the literature review method to collect literature related to recreational sports well-being. After integrating and organizing relevant literature, a drafted 50-item questionnaire was drafted. 11 scholars were invited to examine the content validity of the drafted questionnaire. The criteria for choosing an expert are as follows: (1). An expert is an assistant professor (or above) specialized in the field of sport-related field, currently teaching in a college/university; (2). An expert is an individual who engages in recreational sports. After expert validity examination, similar items were combined, modified, and deleted, and 30 items were kept to be the items of the formal recreational sport well-being scale.

An item analysis was then carried out with the samples of Group 1 to validate the 30-item recreational sport well-being scale. The t-test results showed that the t-value of the 30 items on the scale are all above 3 and are statistically significant (*p* < 0.01), so all the 30 items have discrimination. In terms of the correlation coefficient, the correlation coefficient between each item and the overall scale is all above 0.30 and is statistically significant (*p* < 0.05), which means each item is homogenous to the scale. To sum up, the 30 items mentioned above can all be kept.

### 3.3. Stage 2 Exploratory Factor Analysis, EFA

This study chose Group 1 as the sample for EFA. Through Kaiser–Meyer–Olkin measuring of sampling adequacy, the KMO value is 0.91, close to 1. As to Bartlett’s test of sphericity, the approximate chi-square is 13,166.32. The degree of freedom is 0.91. With a significance of 0.00, it is statistically significant (*p* < 0.05). Therefore, the scale is suitable for factor analysis. The Kaiser-–Meyer–Olkin test result is shown in Table 2.

An EFA was conducted for the 30 items. According to the results, items whose factor loading is below 0.50 and cross-factor items, 16 in all, were deleted. Fourteen items were kept and four factors were extracted. The cumulative variance explained is 78.812%. The EFA result is shown in Table 3.

From Table 3, the reliability result shows that the Cronbach’s α of the overall recreational sport well-being scale is 0.92, which indicates its high reliability. As for all the subscales, the Cronbach’s α of “physical and mental health” is 0.88, “life satisfaction” 0.87, “family flourishing” 0.91, and “positive emotion” 0.92. All the subscales are highly reliable.

### 3.4. Stage 3 Confirmatory Factor Analysis, CFA

After EFA, the recreational sport well-being scale gained a model that has 14 items and 4 factors. A CFA was then conducted with Group 2 samples to analyze scale items’ test for normality, overall model fit test, composite reliability, convergent validity, and discriminant validity.

(1)Test for Normality

Generally, all the scale items will be tested for normal distribution. When an item is against normal distribution, it will affect the model’s estimated correctness. If the skewness absolute value is below 3 and the kurtosis absolute value is below 8, then the scale meets the requirements of normality. For all the items recreational sport well-being scale, the skewness value is between –0.56 to 1.04, and the kurtosis value is between −0.08 to −1.48. The statistics show that all 14 items meet the requirements of normality, as shown in Table 4.

(2)Goodness of Fit Test

The CFA results indicate that the four latent variables found in stage 1 are physical and mental health, life satisfaction, family flourishing, and positive emotion. Furthermore, in these latent variables, physical and mental health and life satisfaction have 4 observed variables, respectively; both family flourishing and positive emotion have 3 observed variables in them. The analysis model of the recreational sport well-being scale is shown in Figure 1.

The result of the goodness of fit model indicates that the χ^2^ of this model is 200.59, *p* = 0.000, which is statistically significant. The χ^2^ is likely to be affected by a large population to be statistically significant; therefore, for the model of recreational sport well-being scale, this study referred to other goodness of fit indexes to judge the overall goodness of fit. The indexes show that SRMR = 0.02, GFI = 0.98, AGFI = 0.97, RMSEA = 0.04, TLI = 0.99, NFI = 0.99, CFI = 0.99, IFI = 0.99, RFI = 0.98, PNFI = 0.77, and PGFI = 0.66. All the indexes passed the required threshold, which means that the recreational sport well-being scale has high goodness of fit, as shown in Table 5.

(3)Composite Reliability

In the recreational sport well-being scale, the composite reliability of the four constructs is between 0.86 and 0.92. All are above 0.70, which indicates the four constructs have good reliability, as shown in Table 6.

(4)Convergent Validity

The factor loadings of the 14 observed variables are between 0.70 and 0.93. All are above 0.50 and are statistically significant, which indicates that the model of recreational sport well-being scale has high convergent validity, as shown in Table 6.

(5)Discriminant Validity

From Table 6, it is clear that the AVE of the 4 factors in recreational sport well-being scale is all above 0.50 (The AVE of life satisfaction is 0.70, physical and mental health 0.69, positive emotion 0.89, and family flourishing 0.51). Furthermore, the square root of each factor’s AVE is also higher than its covariance with other factors in the scale. Therefore, the recreational sport well-being scale has good discriminant validity, as shown in Table 7.

### 3.5. Stage 4 Cross-Validation

The CFA result showed that recreational sport well-being belongs to a first-order, 4-factor oblique model. There are 14 observed variables in this model. To understand whether the model is still valid for different samples, a multiple-group-invariant test was carried out with Group 2 and Group 3 as samples. According to the literature, common tests include configural invariance, metric invariance, covariance invariance, and error variance invariance [33]. Based on Table 8, among all goodness of fit indexes, only the *x*^2^ value is not statistically significant, it is because the Chi value is likely to be influenced by a large population. The rest of them are statistically significant (*p* < 0.05).

The other goodness of fit indexes, such as SRMR, GFI, AGFI, RMSEA, NFI, TLI, CFI, IFI, RFI, PNFI, PGFI, and *x*^2^/df all passed advised thresholds, which suggests that both Group 2 and Group 3′s model have high goodness of fit. In other words, the recreational sport well-being model passed configural invariance.

As for metric invariance, according to Table 8, the Δ*x*^2^ between model A and model B is 1.38, and Δdf is 5, which means it is not statistically significant (*p* > 0.05). The statistics suggested that Group 2 and Group 3 have the same factor leadings, so the model has metric invariance [34]. In terms of covariance invariance, the Δ*x*
^2^ between model 2 and model 3 is 1.54 and the Δdf is 6, which means it is not statistically significant (*p* > 0.05). That is, group 2 and group 3 have the same structural coefficient and passed the covariance invariance test. Last but not least, as for error covariance invariance, the Δ*x*^2^ between model C and model D is 5.50 and the Δdf is 2, which means it is not statistically significant (*p* > 0.05). From the statistics, Group 2 and Group 3 have measurement residual and passed the error variance invariance.

To sum up, recreational sport well-being passed statistical tests such as configural invariance, metric invariance, covariance invariance, and error variance invariance. It means the recreational sport well-being scale can apply to another set of samples; as a result, the recreational sport well-being model passed the cross-validation test, as shown in Table 9.

### 3.6. Stage 5 Nomological Validity

2700 samples from Group 2 and Group 3 were employed to conduct a nomological validity test and the result suggested that there is a positive correlation between recreational sport well-being (r = 0.70, *p* < 0.01). Based on this, it is clear that recreational sport well-being is empirically supported by highly correlated data, which means recreational sport well-being has high nomological validity.

Stage 6 Test-retest method

Group 4, which was composed of 40 participants, filled in the questionnaire twice, and the interval between two tests was two weeks. Through Pearson product-moment correlation analysis, the result (r = 0.99, *p* < 0.01) showed that the recreational sport well-being scale has good stability.

## 4. Discussion

Four factors were developed for the recreational sport well-being scale in this study. The four factors are physical and mental health, life satisfaction, family flourishing, and positive emotions. Life satisfaction and positive emotion are similar to concepts previously proposed by scholars, such as subjective well-being, emotional well-being, and psychological well-being [3,11]. These concepts were often cited by scholars to measure the well-being of taking part in recreational sports. This study found that participation in recreational sports benefits physical and mental health, and the feeling of family flourishing, similar to the concept of consummate well-being proposed by previous studies [12,13], should be taken into consideration. Through validation, this study found that the recreational sport well-being scale can reflect and measure well-being from different perspectives—subjective, emotional, psychological, and flourishing.

Physical and mental health is similar to measurement factors in previous studies [13]. Exercise can relieve pain [35]. Five to ten min of exercise can stimulate the liver to release Insulin-like growth factor 1 into our muscle and central nervous system and facilitate growth hormone to work and exercising for one hour can raise enhance the nervous activities in our body and relieve pain and fatigue [36]. Exercise can improve obesity [37] and is helpful for weight control and reduce the formation of body fat and make us agile [38]. Exercise can prevent osteoporosis since it has a positive effect on bone mineral mass (BMC) [39]. Physical activities with weight are necessary for the development and maintenance of bones. Appropriate weight and resistance training can train muscle strength effectively and slow down bone mineral loss [40]. Exercise can improve sleeping quality as a non-medical method [41,42]. Maintaining a higher physical activity amount and lengthening the time spent on exercise can effectively prevent insomnia [43]. Recreational sport can relieve physical pain, improve obesity, prevent osteoporosis and enhance sleeping quality, gaining a recreational sport well-being due to acquired physical and mental health.

The factor “life satisfaction” is similar to measurement factors in previous studies [6,13]. Regular exercise can reduce stress [44]. When exercising, an individual’s needs to focus on movement and breath; meanwhile, the stress the body feels will reduce, and a stress-eliminating mechanism will be thus generated. Exercise can make an individual find life fulfilling [45]. The key to the benefits of exercise consists in that it must be on a regular basis. The effect is dependent on regular exercise. Making exercise an indispensable part of life will make life fulfilling [46]. Exercise can make life full of energy and it can raise life quality [47]. Studies pointed out that exercise has moderating and mediating effects on life satisfaction. Moderating effect means that exercise can increase positive subjective judgments and life satisfaction directly through health perception and emotional change. The mediating effect suggests that exercise can indirectly improve enjoyment perception, interpersonal relationships, stress and depression, and further increase life satisfaction [48]. Taking part in recreational activities can relieve life pressure and make one find life full of energy and fulfilling, so that life quality will be raised, and recreational sport well-being can also be gained.

The factor “family flourishing” is similar to measurement factors in previous studies [12,13]. Exercise has positive functions and value for family relationships. Participating in dynamic recreational activities will make family members closer to each other and improve the family atmosphere [49]. Previous well-being studies rarely investigated family flourishing from the perspective of exercise. However, an individual’s exercise behavior and experience can become common conversation topics and is helpful for interaction, communication, and discussion among family members. Families can share the pleasant feeling of participating in recreational sports, create an opportunity for family members to exercise together, and develop a warm and harmonious family atmosphere. Taking part in recreational sports is helpful for good family interaction and establishing a harmonious family atmosphere, thus gaining recreational sports well-being of family flourishing.

The factor “positive emotion” is similar to measurement factors in previous studies [3,11]. The higher level of sports participation indicates the more likeliness an individual can gain the feeling of happiness. Positive emotions such as happiness, pleasure, and self-confidence will be more obvious [50]. Exercise can bring an individual joy. When exercising, the chemicals the brain secretes and the neurological media can make one feel mentally pleasant and happy [21]. Exercise will bring positive feelings, which facilitate positive functioning in life. It is clear that taking part in recreational sports will make one feel joyful, pleasant, and happy and gain the recreational sport well-being of positive emotions.

In conclusion, recreational sports well-being refers to the feel degree of physical and mental health, life satisfaction, positive emotions, and a flourishing family after recreational sports. Four factors were developed for the recreational sport well-being scale in this study. The four factors are physical and mental health, life satisfaction, family flourishing, and positive emotions. The merit of this study focuses on the degree of well-being perceived by recreational sports participants and found that measuring recreational sports well-being includes two factors: physical and mental health and family flourishing. Previous studies were unable to measure these concepts; therefore, it is an innovation and breakthrough in this research field. In addition, this study also found that the previous well-being scale can only be used to measure well-being without proposing concrete improvement strategies. This study found that if an individual can only feel low-level well-being when engaging in recreational sports, it means that they need to choose another sport more suitable for them. Long-term regular exercise in daily life can effectively raise well-being.

## 5. Conclusions

The development and validation of recreational sport well-being scale in this study went through 6 stages: (1) the 50-item questionnaire draft, (2) the 30-item formal questionnaire after content validity examined by 11 scholars, (3) EFA: the 14-item, 4-factor (life satisfaction, physical and mental health, family flourishing and positive emotion) recreational sport well-being scale, (4) CFA: in goodness of fit test, 11 indexes (SRMR, GFI, AGFI, RMSEA, TLI, NFI, CFI, IFI, RFI, PNFI and PGFI) suggested the scale model has high goodness of fit; with a compositive reliability between 0.86 and 0.92, the scale has high reliability; the AVE of the 4 factors are above 0.5, so the scale has good convergent validity; in cross-validation tests, the scale passed configural invariance, metric invariance, covariance invariance and error variance invariance, so the scale has cross validity; (5) nomologic validity analysis: a positive correlation between recreational sport well-being and perceived health (r = 0.70, *p* < 0.01), so the scale has nomologic validation; (6) Test-retest reliability: a high positive correlation r = 0.99, *p* < 0.01) reveals the reliability of the scale after a two-week interval. Based on the analysis result mentioned above throughout the six stages, the recreational sport well-being scale developed and validated in this study is highly reliable and valid and can be applied to measure the well-being of various types of recreational sport participants.

## 6. Implications, Limitations, and Suggestions for Future Research

### 6.1. Implication

As the scale was developed exclusively for recreational sport-induced well-being, local governments can use this scale and regularly sample citizens to investigate their recreational sport well-being, and the results can be taken as a reference to increase public well-being.

### 6.2. Limitation

This study did not sample people under the age of 20 since it applied for IRB regulations. It is advisable that future studies can expand the age range to validate the scale structure. In addition, this study did not sample people under the age of 20 since it applied for IRB regulations. It is advisable that future studies can expand the age range to validate the scale structure.

### 6.3. Future Research

It is advisable that future studies can explore the antecedent variables of recreational sports well-being, including the degree of sports participation, flow experience, sports instruction, sports facilities and service quality, and so on. Significant antecedent variables can be taken as strategies for raising recreational sports well-being in the future.

## Figures and Tables

**Figure 1 ijerph-19-08764-f001:**
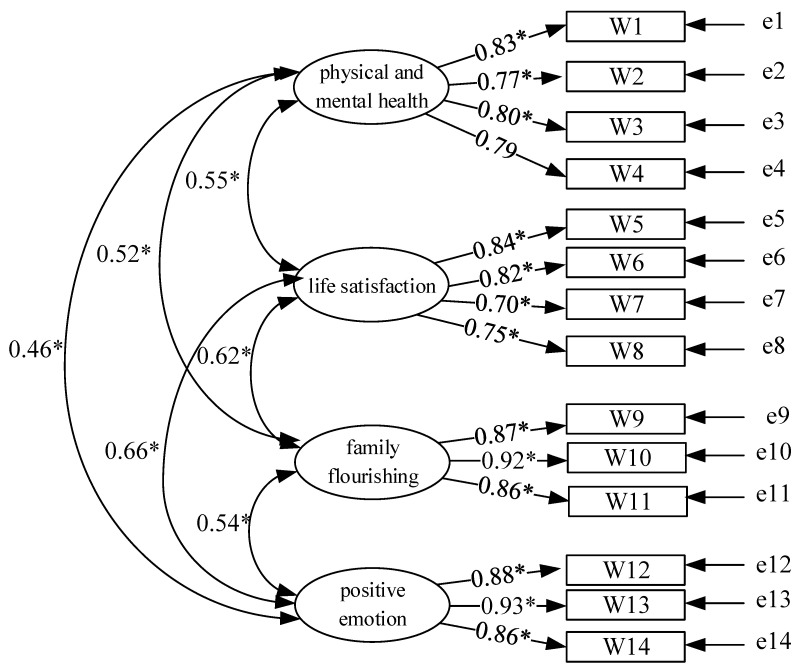
Recreational Sport Well-being Scale’s CFA Analysis Model. * *p* < 0.05.

**Table 1 ijerph-19-08764-t001:** Demographic Information of Research Samples.

		Group 1 (n = 1350)		Group 2 (n = 1350)		Group 3 (n = 1350)		Group 4 (n = 40)		Total (n = 4090)	
		N	%	N	%	N	%	N	%	N	%
Gender	Male	675	50.0	688	51.0	644	47.5	21	50.0	2028	49.5
	Female	675	50.0	662	49.0	706	52.3	19	50.0	2062	50.4
Age	20–25	508	37.6	432	32.0	349	25.9	5	12.5	1294	31.6
	26–50	686	50.8	686	50.8	724	53.6	25	62.5	2121	51.8
	51–64	134	9.9	193	14.3	222	16.4	6	15.0	555	13.5
	Above 65	22	1.6	39	2.9	55	4.1	4	10.0	120	2.9
Marital status	Unmarried	825	61.1	827	61.3	725	53.7	16	40.0	2393	58.5
	Married	489	36.2	486	36.0	576	42.7	22	55.0	1573	38.4
	Single (divorced/widowed)	36	2.7	37	2.7	49	3.6	2	5.0	124	3.0
Education level	High school or below College/	286	21.2	297	22.0	303	22.4	4	10.0	890	21.7
		909	67.3	878	65.0	908	67.3	6	15.0	2701	66.0
		155	11.5	175	13.0	139	10.3	30	75.0	499	12.2

**Table 2 ijerph-19-08764-t002:** Kaiser–Meyer–Olkin Test Analysis.

Dimension	KMO Value	Bartlett Test Value	Degree of Freedom	Significance
Recreational sport well-being	0.91	13,166.32	91	0.00

**Table 3 ijerph-19-08764-t003:** Recreational sport well-being scale EFA and reliability analysis.

Item	Physical and Mental Health	Life Satisfaction	Family Flourishing	Positive Emotion
w1. Relieve physical pain	0.84			
w2. Reduce obesity	0.81			
w3. Prevent osteoporosis	0.81			
w4. Enhance sleeping quality	0.79			
w5. Improve quality of life		0.79		
w6. Live energetically		0.76		
w7. Relieve stress in life		0.74		
w8. Lead a full life		0.71		
w9. Live happily with family members			0.86	
w10. Increase family interactions			0.85	
w11. Enhance family harmony			0.84	
w12. Feel pleasant				0.85
w13. Feel joyful				0.83
w14. Feel happy				0.81
Eigenvalue	7.00	1.79	1.32	1.04
Variance	50.03	12.79	9.40	5.99
Cumulative variance explained	50.03	62.81	72.22	78.21
Subscale reliability	0.88	0.87	0.91	0.92
Overall scale reliability	0.92			

**Table 4 ijerph-19-08764-t004:** The Observed Variable’s Skewness and Kurtosis in Recreational Sport Well-being Scale.

Item	Skewness	Kurtosis
w1	−0.69	0.06
w2	−0.78	0.31
w3	−0.67	0.28
w4	−0.86	0.59
w5	−0.91	0.95
w6	−0.92	0.92
w7	−1.00	0.99
w8	−0.84	0.74
w9	−0.56	−0.08
w10	−0.69	0.02
w11	−0.57	−0.04
w12	−1.04	1.48
w13	−0.98	1.20
w14	−0.91	1.08

**Table 5 ijerph-19-08764-t005:** The Goodness of Fit Indexes of Recreational Sport Well-being Scale.

Indexes	Standard	Group 2 (N = 1350)	Judgment
Model absolute fit measures			
χ^2^ value	χ^2^ value should be as low as possible	200.59	
df		71	
*p*-value	≥0.05	0.00	No
SRMR	<0.80	0.02	Yes
GFI	>0.90	0.98	Yes
AGFI	>0.90	0.97	Yes
RMSEA	<0.08	0.04	Yes
Comparative fit index			
NFI	≥0.90	0.99	Yes
TLI	≥0.90	0.99	Yes
CFI	>0.95	0.99	Yes
IFI	>0.90	0.99	Yes
RFI	>0.90	0.98	Yes
Parsimonious goodness-fit-index			
PNFI	>0.50	0.77	Yes
PGFI	>0.50	0.66	Yes
χ^2^/df	Between 1 and 3	2.82	Yes

**Table 6 ijerph-19-08764-t006:** Test results of the recreational sport’s well-being scale’s reliability and convergent validity.

Factor	Item	Factor Loading	Composite Reliability	Average Variance Extracted
Physical and mental health	w1	0.83	0.88	0.69
	w2	0.77		
	w3	0.80		
	w4	0.79		
Life satisfaction	w5	0.84	0.86	0.70
	w6	0.82		
	w7	0.70		
	w8	0.75		
Family flourishing	w9	0.87	0.91	0.81
	w10	0.92		
	w11	0.86		
Positive emotion	w12	0.88	0.92	0.89
	w13	0.93		
	w14	0.86		

**Table 7 ijerph-19-08764-t007:** Discriminant Validity.

	Life Satisfaction	Physical and Mental Health	Positive Emotion	Family Flourishing
Life satisfaction	0.70			
Physical and mental health	0.19	0.69		
Positive emotion	0.31	0.25	0.89	
Family flourishing	0.21	0.20	0.14	0.81

Note: The diagonal is the AVE, and the rest are the square of the correlation coefficient.

**Table 8 ijerph-19-08764-t008:** Recreational Sport Well-being Model Goodness of Fit Indexes: Group 2 and Group 3 as Samples.

Index	Standard	Group 2 (N = 1350)	Judgment	Group 3 (N = 1350)	Judgment
Model absolute fit measures					
*x^2^* value	*x^2^* value should be as low as possible	200.58		188.37	
df		71		71	
*p*-value	≥0.05	0.00	No	0.00	No
SRMR	<0.80	0.02	Yes	0.02	Yes
GFI	>0.90	0.98	Yes	0.94	Yes
AGFI	>0.90	0.97	Yes	0.97	Yes
RMSEA	<0.08	0.04	Yes	0.04	Yes
Comparative fit index					
NFI	≥0.90	0.99	Yes	0.93	Yes
TLI	≥0.90	0.99	Yes	0.93	Yes
CFI	>0.95	0.99	Yes	0.96	Yes
IFI	>0.90	0.99	Yes	0.99	Yes
RFI	>0.90	0.98	Yes	0.95	Yes
Parsimonious goodness-fit-index					
PNFI	>0.50	0.77	Yes	0.68	Yes
PGFI	>0.50	0.66	Yes	0.70	Yes
*x*^2^/df	Between 1 and 3	2.83	Yes	2.65	Yes

**Table 9 ijerph-19-08764-t009:** Recreational Sport Well-being Model Cross-Validation Result.

Measurement Model	*x^2^* Value	df	*p*-Value	RMSEA	NFI	CFI	ECVI (0.90 Confidence Interval)
Model A	388.95	142	0.00	0.04	0.96	0.97	1.10 (1.06~1.17)
Model B	390.34	147	0.00	0.04	0.95	0.97	1.09 (1.07~1.12)
Model C	391.88	153	0.00	0.04	0.95	0.97	1.07 (1.04~1.08)
Model D	397.37	155	0.00	0.04	0.95	0.97	1.07 (1.03~1.10)
Model comparison	Δ*x^2^*	Δdf			ΔNFI	ΔCFI	
B-A lax	1.38	5	*p* > 0.05		0.00	0.00	
C-B	1.54	6	*p* > 0.05		0.00	0.00	
D-C strict	5.50	2	*p* > 0.05		0.00	0.00	

Ps. Model A: unrestricted model; Model B: factor loading equivalence model; Model C: structural coefficient equivalence model; Model D: measurement residual model.

## Data Availability

Data can be provided upon request.

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
