# Peer review of "Development and Validation of Recreational Sport Well-Being Scale"

_ijerph, 2022, doi:10.3390/ijerph19148764_

Round 1

Reviewer 1 Report

Congratulations on the idea for the article, the aim of which is to develop the "Recreational Sport Wellbeing Scale". The introduction to the article is well-documented, with the psychological aspects of well-being combined. The methodology is described correctly, but requires an explanation why only 40 participants took part in the second stage of the research.
Results: I lack information on the differences in outcomes for the 9 types of recreational sports - can the authors present conclusions on which type of recreational sports provides the "greatest well-being"?

It is very good that the authors presented the chapter "limitations" (called in the work "Suggestions". I think it would be worthwhile for the authors to describe the main disciplines of recreational sports of Taiwanese in two or three sentences, it will allow for a greater understanding of the results among readers from countries where recreational sports are e.g. running, tracking, Nordic walking, etc.

All in all, the article is very interesting, and the corrections that need to be made are intended to enhance the value of the article. When making corrections, it is worth adding two or three articles that were published in recent years (2020-2022) (authors tend to use archival materials).

Author Response

Dear reviewer:

   Thank you for the suggestions. We really appreciate your effort in helping us with the revision. In the following, we will provide responses to your comments respectively.  

Reviewer 1 Comments and Suggestions for Authors

Point 1.

Congratulations on the idea for the article, the aim of which is to develop the "Recreational Sport Wellbeing Scale". The introduction to the article is well-documented, with the psychological aspects of well-being combined. The methodology is described correctly, but requires an explanation why only 40 participants took part in the second stage of the research.

Response 1.

Thanks for the comments.

In the sixth stage, the author mainly considers the test reliability of 9 types of recreational sports, that need to subjects, and try to have subjects in gender, age, marriage, and education level.

Point 2.
Results: I lack information on the differences in outcomes for the 9 types of recreational sports - can the authors present conclusions on which type of recreational sports provides the "greatest well-being"?
Response 2.

Thanks for the comments.

This article does not compare the differences in well-being among 9 types of recreational sports. Further comparative analysis will be in the future.

Point 3.
It is very good that the authors presented the chapter "limitations" (called in the work "Suggestions". I think it would be worthwhile for the authors to describe the main disciplines of recreational sports of Taiwanese in two or three sentences, it will allow for a greater understanding of the results among readers from countries where recreational sports are e.g. running, tracking, Nordic walking, etc.
Response 3.

Thanks for the comments.

Point 4.
All in all, the article is very interesting, and the corrections that need to be made are intended to enhance the value of the article. When making corrections, it is worth adding two or three articles that were published in recent years (2020-2022) (authors tend to use archival materials).

Response 4.

Thanks for the comments.

Updates of recent documents in lines 96-97, ” Regular exercise can enhance well-being[18].”References are also updated in 494-495, “Chang, C. M., Chen, Y. T., Tu, J. B., Chang, S. P.(2022). A study on the relationship among sport constraints, social support and well-being of civil servants in Central Taiwan. Journal of Sport and Recreation Research, 16(3), 91-110. DOI: 10.29423/JSRR.202203 16(3)8.”

Reviewer 2 Report

Dear authors, 

Although the scientific protocol can be considered valid and appropriate for this study, there are 2 main issues to be solved. Firstly, significant references are missing, especially when introducing the thesis statement of the paper. For example, @L116-118 references are needed when you make statements regarding studies and surveys, @L118-120 ("However, in the field of sports, well-being studies always 118 adopted subjective well-being scale as research tool. Whether there are other factors influencing the content of well-being is worth investigation.") a more in-depth analysis should be carried out to explain the importance of other factors that are not included in already validated well-being questionnaires, such as WHO-5, WEMWBS, Hooper's Index.

Secondly, substantial text editing should be carried on, especially within the Introduction and Discussion sections. In general, an English language style should be reviewed in order to improve readability and to integrate short statements or small sentences into the discussion. For example, @L96-97 these sentences should be integrated into a more readable sentence, @L124: "a" article is missing just before "recreational sport well-being scale.”, @L368-381 this paragraph is more likely to a list of redundant sentences than an integrated discussion about the pros and cons of the physical exercise. Moreover, @L368 you report "several studies" to be similar with your findings but at the end you only cite one study. A more in depth discussion is needed when comparing your results with previous studies., @L388-389 "Exercise can make life full of energy. Exercise can raise life quality" could be substituted with "Exercise can make life full of energy and it can raise life quality"., @L419 "the concepts" should be substituted with "these concepts".

Author Response

In response to reviewers’ comments

Dear reviewer:

   Thank you for the suggestions. We really appreciate your effort in helping us with the revision. In the following, we will provide responses to your comments respectively.  

Point 1.

Although the scientific protocol can be considered valid and appropriate for this study, there are 2 main issues to be solved. Firstly, significant references are missing, especially when introducing the thesis statement of the paper. For example,

@L116-118 references are needed when you make statements regarding studies and surveys,

Response 1.

Thanks for the comments.

References are from line 491. “Helliwell, J.; Layard, R.; Sachs, J. World well-being report 2017. New York: Sustainable Development Solutions Network, 2017.”

Point 2.

@L118-120 ("However, in the field of sports, well-being studies always 118 adopted subjective well-being scale as research tool. Whether there are other factors influencing the content of well-being is worth investigation.") a more in-depth analysis should be carried out to explain the importance of other factors that are not included in already validated well-being questionnaires, such as WHO-5, WEMWBS, Hooper's Index.

Response 2.

Thanks for the comments. The study was focuses on the empirical research on the well-being obtained after exercise in the literature, which includes subjective, psychological, social, emotional, and flourishing which are all well-being obtained after exercise.

Point 3.

Secondly, substantial text editing should be carried on, especially within the Introduction and Discussion sections. In general, an English language style should be reviewed in order to improve readability and to integrate short statements or small sentences into the discussion. For example, @L96-97 these sentences should be integrated into a more readable sentence,

Response 3.

Thanks for the comments. Corrected on lines 96-97, “Regular exercise not only enhance well-being, but also is the best predicting for well-being[17.18].”

Point 4.

@L124: "a" article is missing just before "recreational sport well-being scale.”,

Response 4.

Thanks for the comments. Corrected on lines 124.

Point 5.

@L368-381 this paragraph is more likely to a list of redundant sentences than an integrated discussion about the pros and cons of the physical exercise. Moreover, @L368 you report "several studies" to be similar with your findings but at the end you only cite one study. A more in depth discussion is needed when comparing your results with previous studies.

Response 5.

Thanks for the comments. The discussion part is that the results of this study echo the results of previous scholars' research, and the main purpose is to make the connotation of each topic of the research results supported by literature.

Point 6.

@L388-389 "Exercise can make life full of energy. Exercise can raise life quality" could be substituted with "Exercise can make life full of energy and it can raise life quality".,

Response 6.

Thanks for the comments. Corrected on lines 388. "Exercise can make life full of energy and it can raise life quality".

Point 7.

@L419 "the concepts" should be substituted with "these concepts".

Response 7.

Thanks for the comments. Corrected on lines 419.

Reviewer 3 Report

The topic is relevant. In relation to the purposes of the journal, it would be appropriate to revise the introduction underlining the social reasons and referring to public health. in other words, the authors should deeply analyse the public health implications in Taiwan related to relevance of the well-being. Then, the state of the art of well-being should be presented with a more accurate selection of model and instruments available.
In my opinion the authrs could explore the relevance of a cross-cultural perspective in order to justify the development of a new scale.
Regarding to the studies, the authors should provide more references on the statistical methodology. Is Hu and Butler the work that they used to validate the criteria of the factor analysis? What is the statistacal package that they used for the analysis? In relations to the samples caractheristics: the education explained in the table is not enough: how did the authors checked the language reading ability? How did the authors measured the socio-economic status of parrticipants? The information on the Ethic board evalutaion must be integrated.
S
I think that the authors should be consider again the presentation of the results . There are redundancy between the tables and the txts, and there are some tables that could be replaced, for example table 2. I'd expect a comparative table for the variaous sample with the meas, standard deviations, internal consisitency reliability and correlations. I would expect also an external validation with another analogous scale.

Author Response

Dear reviewer:

   Thank you for the suggestions. We really appreciate your effort in helping us with the revision. In the following, we will provide responses to your comments respectively.  

Point 1.

The topic is relevant. In relation to the purposes of the journal, it would be appropriate to revise the introduction underlining the social reasons and referring to public health. in other words, the authors should deeply analyse the public health implications in Taiwan related to relevance of the well-being. Then, the state of the art of well-being should be presented with a more accurate selection of model and instruments available.
In my opinion the authrs could explore the relevance of a cross-cultural perspective in order to justify the development of a new scale.
Regarding to the studies, the authors should provide more references on the statistical methodology. Is Hu and Butler the work that they used to validate the criteria of the factor analysis? What is the statistacal package that they used for the analysis? In relations to the samples caractheristics: the education explained in the table is not enough: how did the authors checked the language reading ability? How did the authors measured the socio-economic status of parrticipants? The information on the Ethic board evalutaion must be integrated.

Response 1.

Thank you for the comments. We had revised the introduction section to stress the importance of the well-being to public health and explain why we need to develop the recreational well-being scale. The reason is because exercise is the best predicting factors according to previous studies. However, when we come to discuss the relationship between exercise and well-being, we can only use general well-being scale. Therefore, in order to measure people’s satisfaction of exercise/recreational exercise, we found it necessary to develop the recreational sport well-being scale.  

Point 2.

I think that the authors should be consider again the presentation of the results . There are redundancy between the tables and the txts, and there are some tables that could be replaced, for example table 2. I'd expect a comparative table for the various samples with the mean, standard deviations, internal consistency reliability and correlations. I would expect also an external validation with another analogous scale.

Response 2.

Thank you for the comments. Table 2 is commonly presented to explain whether the data set is suitable for further factor analysis. If data sets fail to meet the requirements the KMO and Bartlett Spherical test, further EFA test cannot be performed. As for internal consistency, the values were presented in the consistency tests among EFA and CFA analyses. As for correlations, the values were presented among the validity analyses.

Reviewer 4 Report

Specific comments

Introduction

It is important to state that there is a lack of consensus about the meaning of we-being and that there is an extensive usage of different labels etc. Clear philosophical and theoretical foundations of well-being is missing.

The first paragraph (lines 30-44) should be deleted.

Line 57: add approach after hedonic.

Line 61: Change Psychological well-being to Eudaimonic well-being (EWB)

Lines 67-95: There is no flow between the paragraphs.

Lines 87-95: How this paragraph is related with the previous paragraphs?

Lines 96-112: Which type of well-being are you talking about?

Lines 118-119: There are other studies that have used eudaimonic well-being.

Lines 120-124: There is not strong and clear rationale for the study.

It is important to mention studies regarding well-being in sport and recreational well-being.

How recreational well-being is defined?

Materials and methods-Results

The whole part should be restructured. All the information regarding each stage should be separately.

Line 241: Group 1 or 2? In line 167 you mention Group 1.

Lines 172-185: Citations are missing

Lines 297-300. In order to evaluate convergent validity, recreational well-being with other similar constructs should be tested.

Lines 345-350:  Correct nomological instead of nomologic. You should describe perceived health. How did you measure it?

Line 352: 40 participants are enough?

Discussion

Line 418: Why 2 factors?

You should provide a table with all the items.

Discussion-Conclusions

Implications are missing.

Better flow between all the paragraphs is needed.

Overall, you need to include more relevant citations regarding your topic.

Author Response

Dear reviewer:

   Thank you for the suggestions. We really appreciate your effort in helping us with the revision. In the following, we will provide responses to your comments respectively.  

Point 1.

Introduction

It is important to state that there is a lack of consensus about the meaning of we-being and that there is an extensive usage of different labels etc. Clear philosophical and theoretical foundations of well-being is missing.

The first paragraph (lines 30-44) should be deleted.

Response 1.

Thanks for the comments. Lines 30-44 have been removed

Point 2.

Line 57: add approach after hedonic.

Response 1.

Thanks for the comments. The way of citing the literature is described, the original text is a literature review article and does not explain how to decide.

Point 3.

Line 61: Change Psychological well-being to Eudaimonic well-being (EWB)

Response 3.

Thanks for the comments. Corrected on lines 61.

Point 4.

Lines 67-95: There is no flow between the paragraphs.

Lines 87-95: How this paragraph is related with the previous paragraphs?

Response 4.

Thanks for the comments. From lines 67-95, organize many factors of well-being in the article, mainly focus on different factors, and search for the connotation of the question item. From lines 87-95, a research report is compiled, emphasizing the research importance of this article.

Point 5.

Lines 96-112: Which type of well-being are you talking about?

Response 5.

Thanks for the comments. In this paragraph, the author mainly discusses the changes of the nervous system after engaging in exercise, and the perception and feelings produced in the human body from the literature collected. Therefore, this research design items and conducts factor analysis to know which kind of well-being the result belongs to.

Point 6.

Lines 118-119: There are other studies that have used eudaimonic well-being.

Response 6.

Thanks for the comments. There is indeed eudaimonic well-being, but the most common elements in definitions of eudaimonia are growth, authenticity, meaning, and excellence. There is no relevant research on sports as an item to measure feelings.

Point 7.

Lines 120-124: There is not strong and clear rationale for the study. It is important to mention studies regarding well-being in sport and recreational well-being. How recreational well-being is defined?

Response 7.

Thanks for the comments. Recreational sports refer to physical activities individuals engage in after work in their free time, which enable them to enjoy the fun of sports, increase physical activities and attain the goal of health. “Recreational sports well-being refers to the feel degree of physical and mental health, life satisfaction, positive emotions, and family flourishing after recreational sports.”

Corrected on lines 124. “Recreational sports well-being refers to the feel degree of physical and mental health, life satisfaction, positive emotions, and family flourishing after recreational sports.”

Point 8.

Materials and methods-Results

The whole part should be restructured. All the information regarding each stage should be separately.

Line 241: Group 1 or 2? In line 167 you mention Group 1.

Response 8.

Thanks for the comments. Corrected on lines 167. It’s Group 1

Point 9.

Lines 172-185: Citations are missing

Response 9.

Thanks for the comments. Corrected on lines 176. Hair, J.F.; Black, W.C.; Babin, B.J.; Anderson, R.E.; Tatham, R.L. Multivariate data analysis (6th ed.).  Upper Saddle River, NJ: Prentice Hall, 2006.

Point 10.

Lines 297-300. In order to evaluate convergent validity, recreational well-being with other similar constructs should be tested.

Response 10.

Thanks for the comments. Tests of reliability and convergent validity were combined together in Table 6 and presented in lines 297-305.  

Point 11.

Lines 345-350:  Correct nomological instead of nomologic. You should describe perceived health. How did you measure it?

Response 11.

Thanks for the comments. Corrected on lines 193. Previous studies revealed that well-being is correlated to perceived health, so perceived health can be taken as the criterion of well-being. The perceived health scale in this study referred to other perceived health scale previously developed, which contains 6 items and can serve as the criterion tool when testing the nomological validity of the study.

Point 12.

Line 352: 40 participants are enough?

Response 12.

Thanks for the comments. The test of retested reliability normally used the same measurements twice to test whether the measurement results are consistent to make sure the measurement tool is reliable. The most commonly used test for retested reliability is to use the Pearson correlation coefficient (r). If r between two survey data sets is greater than 0.7, the reliability of the measurement is good. According to the central limit theorem (wang and Lee, 2020), the suggested sample size should be greater than 30 and in our study,

we had 40 samples. It should be sufficient enough.

Reference:

Wang, C. Y., & Lee, M. S. (2020). The Development of an Identification Scale to Gauge Cyberbullying and Cyber Aggression on Social Media Sites.  Psychological Testing, 67(1), 61-94.

Point 13.

Line 418: Why 2 factors?

Thanks for the comments. Four factors were developed for the recreational sport well-being scale in this study. The four factors are physical and mental health, life satisfaction, family flourishing and positive emotions.

Round 2

Reviewer 2 Report

Dear authors, 

In the latest version the manuscript is improved and it is suitable for publication.

Author Response

Point 1.

In the latest version the manuscript is improved and it is suitable for publication.

Response 1.

Thanks for the comments.

Reviewer 4 Report

Specific comments

·       In terms of the introduction, it is important to state that there is a lack of consensus about the meaning of well-being and that there is an extensive usage of different labels etc. I haven’t seen any changes in the manuscript regarding this comment.

·       Line 57: add the word “approach” or perspective” after “ Subjective well-being (SWB) is based on hedonic”.

·       Lines 65-67: Eudaimonic well-being is not only measured by these 6 aspects, you should mention other measures as well.

·       Lines 118-119: “However, in the field of sports, well-being studies always adopted subjective well-being scale as research tool”. You need to rephrase this because there are other studies that have used eudaimonic well-being.  You commended: “There is indeed eudaimonic well-being, but the most common elements in definitions of eudaimonia are growth, authenticity, meaning, and excellence. There is no relevant research on sports as an item to measure feelings”. It is important to make clear if your definitions is based on hedonic or/and eudaimonic aspects of well-being and give an explanation for this.

·       Lines 120-124: How recreational well-being is defined? The definition is not addressed here. It is mentioned in lines 419-422.

·       You should provide a table with all the items of your scale.

·       Regarding the Suggestions section, better flow between all the paragraphs is needed.

Author Response

Dear reviewer:

   Thank you for the suggestions. We really appreciate your effort in helping us with the revision. In the following, we will provide responses to your comments respectively.

Point 1.

  • In terms of the introduction, it is important to state that there is a lack of consensus about the meaning of well-being and that there is an extensive usage of different labels etc. I haven’t seen any changes in the manuscript regarding this comment.

Response 1.

Thanks for the comments. We had deleted non-related definition and focus on subjective (hedonic) well-being and Eudaimonia well-being(EWB) and in the end we pointed out the need to develop the recreational sport related well-being scale to measure the SWB and EWB induced exclusively from recreational sport involvement.

Point 2.

  • Line 57: add the word “approach” or perspective” after “ Subjective well-being (SWB) is based on hedonic”.

Response 2.

Thanks for the comments. We had added the word.

Point 3.

  • Lines 65-67: Eudaimonic well-being is not only measured by these 6 aspects, you should mention other measures as well.

Response 3.

Thanks for the comments.

Point 4.

  • Lines 118-119: “However, in the field of sports, well-being studies always adopted subjective well-being scale as research tool”. You need to rephrase this because there are other studies that have used eudaimonic well-being. You commended: “There is indeed eudaimonic well-being, but the most common elements in definitions of eudaimonia are growth, authenticity, meaning, and excellence. There is no relevant research on sports as an item to measure feelings”. It is important to make clear if your definitions is based on hedonic or/and eudaimonic aspects of well-being and give an explanation for this.

Response 4.

Thanks for the comments. We had revised the sentences. Please refer to lines 112-119.

Point 5.

  • Lines 120-124: How recreational well-being is defined? The definition is not addressed here. It is mentioned in lines 419-422.

Response 5.

Thanks for the comments. We had deleted the “recreational well-being” item

Point 6.

  • You should provide a table with all the items of your scale.

Response 6.

Thanks for the comments. We had included the questionnaire in the appendix.

Point 7.

  • Regarding the Suggestions section, better flow between all the paragraphs is needed.

Response .

Thanks for the comments. We had revised the suggestion section.
